# How Can Restaurant Companies Effectively Deliver CSR Efforts to Consumers on Social Media?

**DOI:** 10.3390/bs13030211

**Published:** 2023-03-01

**Authors:** Sae-Mi Lee, Minseong Kim

**Affiliations:** 1School of Global Business, Kyungil University, 50 Gamasil-gil, Hayang-eup, Gyeongsan-si 38428, Gyeongbuk, Republic of Korea; 2Department of Management & Marketing, College of Business, Louisiana State University Shreveport, Shreveport, LA 71115, USA

**Keywords:** source credibility, CSR, social media, CSR communication, restaurant

## Abstract

This study emphasized the source credibility effect on consumers’ responses to restaurant companies’ CSR efforts on social media. Based on the source credibility brand development process (i.e., brand trust, brand affect, and brand engagement), this study employed an online survey-based experiment and revealed the meaningful associations among types of CSR posters (a restaurant brand vs. a nonprofit organization), brand trust, brand affect, and brand engagement. The empirical results indicated that the nonprofit organization has higher significant effects on brand trust and brand affect than those of the restaurant brand. Additionally, in the case of the nonprofit organization, the effects of brand trust and brand affect on brand engagement were significantly stronger than that of the restaurant brand. Based on the empirical findings, this study provided several theoretical and managerial implications for restaurant companies’ CSR communication on social media.

## 1. Introduction

Since the 1990s, corporate social responsibility (CSR) has attracted media attention in the United States, and it has played a role in many corporations’ public relations campaign arsenals [1]. A common assumption is that firms need to initiate socially responsible activities for their businesses as a consequence of ethical obligations, which are both externally and internally encouraged. Scholars and practitioners have supported this notion by suggesting that there are a variety of reasons for acting in a socially responsible way [2]. For instance, as the number of ethically conscious consumers is increasing, CSR initiatives can bring differential advantages (such as enhancement of company reputation and consumer rapport) to corporations [3,4].

Social media platforms have been used by companies as communication tools to convey their CSR initiatives in a manner that is easily accessible to both current and prospective consumers [5,6]. This is because companies can better respond to external demands and pressure from consumers by interacting with the consumers about their CSR efforts [6,7]. However, not all consumers on social media necessarily view the companies as socially responsible [8]. For example, some consumers tend to be skeptical of hospitality companies. In particular, foodservice enterprises that have uploaded posts about new menu development while simultaneously mistreating animals for their menus tend to prompt outrage. This can be seen in examples of social media posts, such as “Stop animal abuse now by boycotting the company X! They refuse to make common sense changes to their animal welfare policy, something that over 85 other corporations have already done! Why is the company X lagging behind when it comes to animal cruelty?” (Source from Facebook in 2018). In addition, a study by Mohr, Webb, and Harris [9], found that while about half of consumers had high levels of favorable attitudes toward companies engaging in CSR initiatives, one-third of individuals considered the companies’ CSR activities as focused on the companies’ self-interest. The study findings indicated that not all consumers have positive attitudes toward companies that implement CSR initiatives. Hence, focusing only on the delivery of CSR efforts may not be enough for consumers to formulate positive attitudes toward companies and/or brands because consumers’ skepticism of firms engaging in CSR initiatives may hinder the firms’ success with the initiatives [10].

How do companies use communication channels to disseminate their CSR activities? Most of the communication channels associated with a firm’s CSR activities are controlled by the firm [11]. According to the existing literature on CSR, however, a trade-off exists between the CSR communications’ controllability and credibility (the more in control the communicator is, the less credible it is) [11]. Specifically, consumers tend to be more critical of a message from a source that is perceived to be self-interested or biased (such as a CSR message generated by a firm), which may be more likely to generate consumers’ skepticism than non-firm sources [12]. When learning about a company’s CSR activities from a non-corporate source, consumers are more likely to react positively than learning CSR activities from a company source [13]. To answer the above question, this study will conceptually establish and empirically test the psychological process underlying the effect of the CSR message sources on hospitality consumers in the context of brand development on social media. Specifically, the objectives of our research are: (1) to empirically test whether a nonprofit organization’s CSR post regarding a foodservice brand’s CSR activity has a stronger impact on brand trust, brand affect, and brand engagement than if the brand was based on the fundamental notion of the source credibility theory; and (2) to rigorously examine the impact of the type of CSR poster on brand engagement via brand trust and brand affect.

The current research contributes to the existing literature on CSR by applying source credibility theory to the hospitality industry and social media consumer behavior contexts, compared to previous empirical studies that did not consider CSR message sources on social media (e.g., who can be a better and more credible CSR communicator in the social media platforms?) [14,15]. In other words, prior empirical studies based on CSR in the hospitality industry have focused primarily on identifying the impact of CSR initiatives on consumers’ psychological and behavioral responses to a brand/company that implements CSR activities, such as perceived value, satisfaction, and loyalty toward the brand/company [16]. However, our research extends source credibility theory to focus on a CSR communicator (i.e., a foodservice brand itself vs. a third party, such as a nonprofit organization) and its impact on brand trust, brand affect, and brand engagement in the context of social media. In addition, from a practical perspective, since our research model predicts social media users’ engagement with a foodservice brand, its empirical findings provide foodservice brand marketers and CSR directors with meaningful implications for enhancements to social media users’ engagement with their brands that are considered to be a powerful communication tool, spreading the foodservice brands’ CSR initiatives without geographic and time limits in the digital world [17]. With this practical implication, marketers and CSR directors at food service brand companies can reduce skepticism about their CSR efforts among both existing and potential customers on social media and maximize the effectiveness and efficiency of their CSR communications via social media platforms [16,17].

The structure of this article consists of: (1) the literature review section demonstrating the concepts of source credibility, brand trust, brand affect, and research hypotheses; (2) the methodology section focusing on experimental design and empirical results; (3) the conclusion section with limitations as well as theoretical and practical implications.

## 2. Literature Review

### 2.1. Source Credibility

Source credibility originates from the influence of a communicator’s positive characteristics on people’s acceptance of a message from the communicator [14]. Source credibility is the “extent to which the source is perceived as possessing expertise relevant to the communication topic and can be trusted to give an objective opinion on the subject” [15] (p. 43). According to source credibility theory, therefore, message effectiveness depends upon the degree to which people view a communicator as being trustworthy and having expertise [18]. Trustworthiness (or believability) of a source function as an indicator of the perceived decency or integrity of the source and thus determines credibility [19]. For example, if individuals perceive a source as being credible, they see the source as far more persuasive than other sources. Expertise refers to how receivers perceive a source as being skilled, knowledgeable, or experienced in the source’s subject field [19]. For instance, if individuals perceive a source to be an expert about a topic, they are more likely to believe the source [20].

Based on source credibility theory, in the CSR context, consumers are also more likely to positively respond to a company’s CSR activities when they perceive that the source delivering the company’s CSR efforts is credible and honest [11]. However, when the company promotes its CSR activities on social media, many users may tend to see the company’s CSR efforts as a business strategy to increase its financial performance rather than a genuine interest in social and environmental issues [11,21]. In addition, consumers may have skepticism about the company’s CSR activities if the company uses company-controlled communication (a company/brand promotes its CSR activities by itself). Thus, companies sometimes employ an organization (or person) for social media public relations strategies for being socially responsible. It is assumed that the right organization needs to be well-skilled and knowledgeable in the social responsibility field to be a credible source [18]. Third parties that are perceived as more credible and knowledgeable than companies in terms of social and/or environmental issues can assist consumers in accepting the companies’ CSR messages in a more positive manner [22]. In spite of the important role of a CSR communicator, prior research on source credibility and CSR has emphasized a type of CSR content, CSR-related language style, and positive and/or negative user comments on social media, and investigated their influences on social media users’ trust formation process [23,24]. To fill the academic gap and extend the theoretical boundary of source credibility theory, this study proposes a different type of CSR where communicators may be able to generate a different level of perception of a brand that develops and implements CSR initiatives and delivers that CSR-related information to social media users.

### 2.2. Brand Trust and Brand Affect

Trust is conceptualized as the involvement in a calculative process based on the estimation of the rewards versus costs of staying in the current relationship beyond the other party’s ability to continue to meet its obligations [25]. Trust is also the inference that a firm or brand will act in its consumers’ best interests based on the two parties’ (firm/brand and consumers) shared values and goals [26,27]. Brand trust is “the willingness of the average consumer to rely on the ability of the brand to perform its stated function” [26] (p. 82). Hence, consumers’ beliefs in honesty, safety, and reliability can all be critical aspects of brand trust, which the consumers incorporate in their operationalization of brand trust [23].

Moorman, Zaltman, and Deshpande [28] defined brand affect as the potential of a brand to elicit consumers’ positive emotional status as a consequence of experiencing the brand. In this study, a hospitality company’s CSR activities may be more closely associated with consumers’ emotional responses to the company/brand than their trust toward the company/brand. Prior research has indicated that individuals’ mental representations tend to be influenced by stimuli’s affective features rather than actual descriptive characteristics of the episode stimuli (such as the company’s CSR activities) [29,30]. Hence, the original input’s affective quality serves as the first emerging attribute when individuals attempt to retrieve a particular object (such as name, story, person, and episode) from memory [31]. Even though it is hard to exclude the actual cognitive (descriptive) characteristics of memory, the studies of Zajonc [31] and Forgas and Bower [29] indicate that brand affect can be important for consumers’ brand recognition, recall, attitudes, and behaviors. Previous empirical studies in CSR have well-investigated how CSR initiatives lead consumers to build brand trust, brand attitudes, and brand loyalty [16,24]. However, prior research has overlooked how CSR initiatives result in consumers’ emotional behavioral responses to a brand although brand affect has been studied as a core component of brand performance and brand equity in the general consumer behavior field [32]. Accordingly, the inclusion of brand affect in our study’s research model may fill the academic gap in the CSR context by considering the social media users’ emotional aspect.

### 2.3. Research Hypotheses

Prior research on CSR communication has indicated that third-party communication is more effective than company-controlled communication [11]. In particular, consumers tend to perceive third parties with expertise in social and environmental issues as more trustworthy than companies—even those that are perceived as socially responsible among consumers [33]. Even though each organization (the third party and the company) delivers the company’s CSR activities at the same time, the third party’s CSR information generates more positive attitudes toward the company and higher behavioral intentions for the company [33,34]. This is because consumers perceive that the third party provides more unbiased information and evaluation of the company’s CSR activities [13]. Therefore:

**H1.** 
*In line with the source credibility framework, a nonprofit organization’s CSR post regarding a brand’s CSR activity will have stronger impacts on brand trust (H1-1), brand affect (H1-2), and brand engagement (H1-3) than those of the brand.*


Many brands have used social media as a rich communication platform for building and maintaining a strong relationship with their consumers (such as trust) [27]. In social media, consumers are able to easily communicate with other users and share their opinions concerning a particular brand, company, or product with others (such as existing and potential consumers and marketers) [35]. Theoretically, this platform helps to develop trust between two parties (such as the brand and the consumers) through repeated interactions [36]. For example, via repeated interactions (such as sharing a meaningful experience with a particular brand or receiving feedback from other users or the brand itself), consumers tend to perceive a brand as more trustworthy through the strengthening of the relationship between the brand and consumers [35]. In social media, consumers with high levels of trust toward a particular brand are more likely to interact with the brand’s videos, pictures, posts, and with other consumers [27].

Prior studies in CSR communication have mostly emphasized CSR communication strategies’ effectiveness based on emotional communication [37]. According to the emotional approach, CSR communication is using “the social marketing concept based on positive emotions and entertaining features, which are more effective than the hitherto predominant fact-oriented style of consumer information” [38] (p. 198). As such, consumers’ particular emotional responses to a brand from its CSR messages have played significant roles in determining brand engagement. Thus:

**H2.** 
*The relationship between the type of CSR poster and brand engagement is mediated by brand trust (H2-1) and brand affect (H2-2).*


## 3. Experiment 1

Experiment 1 investigates the impacts of source credibility on social media consumers’ brand trust, brand affect, and brand engagement. Participants will be exposed to one of two possible posts on social media (offered by either a foodservice brand or nonprofit organization) featuring the brand’s CSR initiatives (“We [or this foodservice brand] have a number of goals and environmental and social performance indicators that we [or this foodservice brand] use to measure progress and hold ourselves [or itself] accountable. We [or this foodservice brand] report annually on our [or its] climate change, energy and greenhouse gas emissions, water management, and forecasts strategies through the Carbon Disclosure Project.”). For the source credibility condition, the two posts describe the same CSR initiatives of the brand. All posts are accompanied by photos of the brand’s CSR reports and manipulated by two types of posters (a brand or nonprofit organization), respectively.

### 3.1. Method

A sample was collected using the crowdsourcing website Amazon’s Mechanical Turk and participants will be awarded $1.00 for their completion of the survey [39]. This study conducted an online survey-based experiment, and all participants were randomly assigned to one of two possible conditions in which they were asked to view a social media message, indicating a brand’s CSR initiatives posted by either the brand or a third party (nonprofit organization). After viewing the post, participants completed a survey that measured their levels of brand trust, brand affect, and brand engagement on social media. Each of the social media posts included manipulations of types of posters.

For manipulation check in source credibility, four Likert-type items (1 = “not at all,” 7 = “very much”) are adapted from West [40]. Participants will be asked to indicate their agreement with the statement: (1) “The information on the post is trustworthy;” (2) “The information on the post is accurate;” (3) “The information on the post is fair;” (4) “The information on the post is unbiased” (α = 0.895). Four Likert-type items and three Likert-type items (1 = “strongly disagree,” 7 = “strongly agree”) are used to, respectively measure brand trust (α = 0.841) and brand affect (α = 0.808) and are adapted from the study of Chaudhuri and Holbrook [26]. Finally, to measure brand engagement, the participants are asked to indicate their willingness to express engagement with the brand (α = 0.863) [41].

### 3.2. Results

Before testing hypotheses, a one-way analysis of variance (ANOVA) was conducted to check that the experimental condition was successfully manipulated (covariate: perceived familiarity with the restaurant brand) (i.e., a brand = 3.171 vs. nonprofit organization = 3.601, *F* [(1, 205] = 18.124, *p* < 0.01). The purpose of a one-way ANOVA is to statistically determine whether the mean value of one group is significantly different from that of another group (i.e., each construct’s mean values of the brand-exposed group vs. each construct’s mean values of the nonprofit organization-exposed group) by controlling for potential effects on a dependent variable with consideration of a covariate (the external influences of perceived familiarity with the foodservice brand on brand trust, brand affect, and brand engagement) [42]. To test H1, one-way ANOVAs were performed to empirically investigate the significant differences between a brand-exposed group and a nonprofit organization-exposed group in brand trust (H1-1), brand affect (H1-2), and brand engagement (H1-3) (covariate: perceived familiarity with the restaurant brand). The empirical findings revealed that there were significant differences between the groups in brand trust (i.e., restaurant brand group = 3.055 vs. nonprofit organization group = 3.406, *F* [(1, 205] = 32.616, *p* < 0.01), brand affect (i.e., restaurant brand group = 3.330 vs. nonprofit organization group = 3.641, *F* [(1, 205] = 25.903, *p* < 0.01), and brand engagement (i.e., restaurant brand group = 3.295 vs. nonprofit organization group = 3.673, *F* [(1, 205] = 29.687, *p* < 0.01), supporting H1. The outcome of one-way ANOVAs in this study confirmed the statistically significant differences between the restaurant brand-exposed group and the nonprofit organization-exposed group in trust toward the foodservice brand, affect to the foodservice brand, and engagement with the foodservice brand.

This study tested H2 by conducting a confirmatory factor analysis (CFA) for reliabilities and validities and structural equation modeling for estimating the proposed paths. The purpose of a CFA is to assess how well each construct is significantly represented by the corresponding indicators, items, or measures, and how well a proposed measurement theory or model is significantly represented by all constructs and respective indicators (i.e., reliabilities and validities in this study) [42]. In other words, via a CFA, scholars in the social science field are able to evaluate the qualities of the proposed measurement theory or model as well as indicators simultaneously from a psychological perspective. Hence, an outcome of a CFA addresses that the proposed theory or model in this study is well-operationalized by a set of constructs and indicators by checking their reliabilities and validities with values of factor loadings (i.e., should be greater than 0.50 for convergent validity), critical ratios (i.e., should be greater than 1.96 for convergent validity), AVE (i.e., should be greater than 0.50 for discriminant validity), and CCR (i.e., should be greater than 0.70 for reliability) before proceeding with structural equation modeling [43]. Table 1 indicates that convergent and discriminant validities of each construct were confirmed via CFA.

After checking the reliabilities and validities of all indicators, this study performed structural equation modeling by assessing the hypothesized associations between the constructs. Compared to other statistical approaches that investigate only a single relationship between an independent variable(s) and a dependent variable(s), structural equation modeling enables scholars in the social science field to allow for multiple relationships among independent variables, mediators, and dependent variables simultaneously by considering the measurement error and validities of a prespecified set of research hypotheses in this study. In other words, structural equation modeling offers the ability to rigorously test the hypothesized key relationships as well as to evaluate the measurement properties in our study, resulting in meaningful contributions to the extant literature in this field [42,43]. The model fit indices were acceptable to interpret the hypothesized paths: χ^2^ = 64.725, d.f. = 39, *p* < 0.01, RMSEA = 0.056, NFI = 0.942, CFI = 0.976, TLI = 0.965. The type of poster (i.e., independent variable) was recoded into dummy variables (i.e., 0 = restaurant brand, 1 = nonprofit organization). First, the nonprofit organization has higher significant effects on brand trust (standardized estimate = 0.243, standardized error = 0.095, critical ratio = 3.315, *p* < 0.01) and brand affect (standardized estimate = 0.224, standardized error = 0.105, critical ratio = 3.002, *p* < 0.01) than those of the restaurant brand. Second, in the case of the nonprofit organization, the effects of brand trust (standardized estimate = 0.276, standardized error = 0.080, critical ratio = 3.482, *p* < 0.01) and brand affect (standardized estimate = 0.425, standardized error = 0.077, critical ratio = 5.102, *p* < 0.01) on brand engagement were significantly stronger than that of the restaurant brand. Lastly, since the direct impact of the independent variable on the dependent variable (i.e., brand engagement) was not statistically significant (standardized estimate = 0.097, standardized error = 0.084, critical ratio = 1.509, *p* > 0.05), brand trust and brand affect served as full mediators in the relationship between the independent variable and the dependent variable (i.e., the standardized indirect effect of the independent variable on the dependent variable = 0.162, *p* < 0.01, lower bounds = 0.080, upper bounds = 0.258; 95% Bias-corrected confidence interval level), supporting H2 (see Figure 1).

## 4. Conclusions

This study focuses on the role of source credibility in developing a strong brand via CSR communication on social media. This study investigates whether or not hospitality consumers differ in the brand development process of CSR communication, depending on who delivers the CSR message. The empirical findings may be useful for understanding the psychological framework of hospitality consumers in which to adapt CSR communication for brand development on social media platforms by maximizing the effectiveness of hospitality companies’ CSR activities. More specifically, although consumers tend not to trust information about a company’s CSR activities generated from the company itself, the consumers may formulate higher levels of brand trust, brand affect, and brand engagement if the CSR efforts are shared by a third party (i.e., in this study, a nonprofit organization).

One of the meaningful theoretical contributions of our research is to refine the relationship among the source of credibility (i.e., the type of CSR poster in this study), brand trust, brand affect, and brand engagement in the contexts of hospitality and social media. The existing hospitality literature on CSR has focused primarily on the impact of CSR initiatives and activities on consumers’ cognitive, emotional, and behavioral responses to a brand that develops and implements CSR activities without consideration of the role of a CSR delivery method [16,44]. However, our empirical research extends source credibility theory by applying it to the CSR and social media contexts via manipulation of the types of a CSR communicator (i.e., restaurant brand itself vs. a third party, such as a nonprofit organization) rather than CSR communication channel choice (i.e., official websites vs. social media platforms) [17,45]. This study provides a new avenue for the hospitality industry, using the critical role of a CSR communicator in building consumers’ brand trust and positive emotions toward hospitality brands within the social media context, based on source credibility theory. In practice, digitalized marketing channels are considered to be and used as powerful and influential marketing tools for delivering a brand’s marketing efforts in an effective and efficient way (i.e., CSR efforts in this study). However, the empirical findings of our research reveal that the effectiveness and efficiency of social media marketing for CSR depends on who delivers a brand’s CSR initiatives to social media users (or existing and potential customers). Therefore, marketers and CSR directors at a foodservice brand should work closely with nonprofit organizations via the establishment of strategic alliances with credible and popular nonprofit organizations that can serve as alternative CSR communicators in the digital world. However, food service brands should consistently check nonprofit organizations’ content and posts that are aligned with the brands’ CSR values.

The second theoretical implication of our study is to conceptually and empirically examine the determinants of brand engagement in the social media context. It is necessary for scholars and practitioners in the hospitality industry to identify and understand the motivational factors affecting social media users’ engagement. However, prior research on CSR in the hospitality industry has focused mainly on investigating the determinants of brand satisfaction, attitudes, and loyalty [16,46]. Most hospitality companies worldwide employ digitalized communication channels, such as official websites and social media channels, to actively interact with their consumers without any geographic or time limits. The empirical findings address that CSR-related brand trust and brand affect serve as core drivers of brand engagement on social media in addition to brand attitudes and loyalty [46,47]. In other words, this study finds that brand engagement can be formed by restaurant brands’ CSR initiatives via only brand trust and brand affect. Practically, these empirical findings confirm that marketers and CSR directors at restaurant brands should emphasize the mediating roles of brand trust and brand affect in building social media users’ brand engagement. Hence, marketers and CSR directors at restaurant brands attempt to create credible CSR content and post those on social media, and try to dissuade consumers’ negative emotional responses to CSR-related social media content and posts by working with nonprofit organizations.

Lastly, our research proposes some directions for future empirical studies in the hospitality industry based on three limitations. First, the outcomes of this study may not be generalized to all types of CSR communication strategies due to a methodological limitation of an experimental design (i.e., celebrity and anthropomorphism or video vs. picture) [48]. Furthermore, our study does not consider the potential impact of the number of followers, other CSR posts, and comments on consumers’ levels of brand trust, brand affect, and brand engagement (e.g., no comment option in the stimuli). For example, positive and/or negative comments on the CSR post may have an impact on consumers’ responses to the CSR posts of the company [49]. Therefore, future research must consider or control the above factors to enhance this study’s experimental design’s external validity. Second, this study does not provide the participants with enough information about a brand’s CSR initiatives (i.e., in addition to the post’s contents, the participants should have access to each CSR report on the post to learn about the restaurant brand’s CSR initiatives and activities in detail). In other words, there is a possibility that a participants’ information processing does not appropriately work due to a lack of information. Therefore, future research should provide participants with enough information about a restaurant brand’s CSR initiatives/activities to explore the impact of source credibility on their CSR information processing in a more detailed manner, which results in brand engagement on social media. Third, this study acknowledges that the stimulus may have been more rigorously designed-controlled. Although this study included participants’ perceived familiarity with the restaurant brand as a control variable, other aspects of the given restaurant brand might have led participants to express different levels of brand trust, brand affect, and brand engagement. Thus, future studies should consider designing more feasible stimuli, reflecting various aspects, to rule out this possible alternative explanation.

## Figures and Tables

**Figure 1 behavsci-13-00211-f001:**
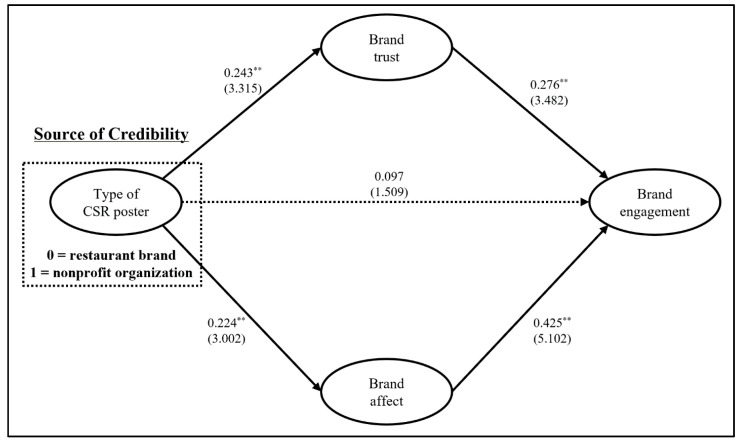
Results of structural equation modeling. Note. Standardized estimate (critical ratio), solid line: significant path, dotted line: insignificant path. ** *p* < 0.01.

**Table 1 behavsci-13-00211-t001:** Results of CFA, AVE, and CCR for experiment 1.

Constructs and Operationalization	Items	Factor Loading	Critical Ratio	AVE	CCR
Brand trust	I trust this restaurant brand.	0.759	Fixed	0.591	0.848
	I rely on this restaurant brand.	0.884	12.618		
	This is an honest restaurant brand.	0.850	12.287		
	This restaurant brand is safe.	0.532	7.437		
Brand affect	This restaurant brand makes me happy.	0.809	Fixed	0.660	0.851
	This restaurant brand gives me pleasure.	0.874	11.715		
	I feel good when I think about this restaurant brand.	0.651	9.337		
Brand engagement	I will share this restaurant brand’s CSR-related video or photos on my social media.	0.800	Fixed	0.686	0.867
	I will use trending words such as hashtags for this restaurant brand’s CSR activities while using social media.	0.900	13.354		
	I will share my opinions about this restaurant brand’s CSR activities with other users of this brand.	0.779	11.913		

χ^2^ = 52.238, d.f. = 32, *p* < 0.05, RMSEA = 0.055, NFI = 0.951, CFI = 0.980, TLI = 0.972. Note. AVE: Average Variance Extracted; CCR: Composite Construct Reliability.

## Data Availability

The data presented in this study are available on request from the corresponding author. The data are not publicly available due to privacy reasons.

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
