# Peer review of "How Can Restaurant Companies Effectively Deliver CSR Efforts to Consumers on Social Media?"

_behavsci, 2023, doi:10.3390/bs13030211_

Round 1
Reviewer 1 Report
I appreciate the opportunity to review your work. I share your interest in this topic, and want to reinforce your efforts.
Intro: To begin, you did not properly motivate this study from a practical standpoint. In light of the Behavioral Sciences' aim and scope, this was a big oversight on your part. Why is it important to examine the CSR Efforts to Consumers on Social Media right now, especially in light of the current competitive conditions? What kinds of direct, industry-specific evidence can you offer that would make industry leaders sit up and pay attention to your study? Answers to these and related questions will help make a much stronger case for pursuing this line of inquiry. Please also write your objectives at the end of the intro part.
Results: In your abstract you mentioned that your study provided both theoretical and practical implications. You did not appear to have given sufficient attention to either theoretical or practical implications. Please reconsider your implications and pay close attention to the aforementioned section.
Author Response
Please see the attached response letter.

Reviewer 2 Report
In my opinion, the publication submitted for review is interesting and worthy of publication, especially because it deals with current topical issues and topics. However, before final approval and referral for publication, I would like to ask the authors to make a few changes to make the article even more scientifically valuable.
(1).The article needs to address more clearly the novelty of the research. In addition, in the introduction section, I recommend to insert the structure of the article. Please explain, how the model / framework was developed, how is different theory used to derive variables in the model. Please also explain the novelty of this study.
(2).You need to go beyond describing a series of relevant references, and tell us how your interpretation of the literature shows the gaps that exist, and how the proposed approach to the literature brings about novel opportunities to reinterpret the literature that will allow an advancement in our understanding in the field. Additionally, more updated literature could have been used.
(3).The description of the methodology is lacking in detail and scope. The paper need a deeper level of analysis of the key issues. How do you explain your statistical analysis outcomes and how do they contribute the current research?
(4).The discussion do not provide enough detail. The theoretical analysis of this paper is limited. More theoretical building needs to be further explicated. It would be better to situate the discussion in the wider literature and relate it to broader debates and issues in global context.
(5).In the final section, I think that it would be valuable to enlarge the discussion about the limits of this work, better discussing possible aspects to refine and deepen in future research.
Author Response

(The authors gave the same response as above.)

Round 2
Reviewer 2 Report
The authors have effectively addresses my comments and remarks and the quality and clarity of the manuscript have been improved.